# Behavioral Reactions to Job Insecurity Climate Perceptions: Exit, Voice, Loyalty, and Neglect

**DOI:** 10.3390/ijerph20095732

**Published:** 2023-05-05

**Authors:** Ümran Yüce-Selvi, Nebi Sümer, Yonca Toker-Gültaş, Lena Låstad, Magnus Sverke

**Affiliations:** 1Department of Psychology, Eskisehir Osmangazi University, 26480 Eskişehir, Türkiye; 2Faculty of Arts and Social Sciences, Sabancı University, 34956 İstanbul, Türkiye; 3Department of Psychology, Middle East Technical University, 06800 Ankara, Türkiye; 4Department of Education, Stockholm University, 106 91 Stockholm, Sweden; 5Department of Psychology, Stockholm University, 106 91 Stockholm, Sweden

**Keywords:** job insecurity climate, consequences, employee behaviors, exit, voice, loyalty, neglect

## Abstract

Past work has extensively documented that job insecurity predicts various work- and health-related outcomes. However, limited research has focused on the potential consequences of perceived job insecurity climate. Our objective was to investigate how the psychological climate about losing a job and valuable job features (quantitative and qualitative job insecurity climate, respectively) relate to employees’ exit, voice, loyalty, and neglect behaviors, and whether such climate perceptions explain additional variance in these behaviors over individual job insecurity. Data were collected through an online survey using a convenience sample of employees working in different organizations in Türkiye (*N* = 245). Hierarchical multiple regression analyses showed that quantitative job insecurity climate was associated with higher levels of loyalty and neglect, while qualitative job insecurity climate was related to higher levels of exit and lower levels of loyalty. Importantly, job insecurity climate explained additional variance over individual job insecurity in exit and loyalty. Our findings underscore the importance of addressing job insecurity in a broader context regarding one’s situation and the psychological collective climate. This study contributes to addressing the knowledge gap concerning job insecurity climate, an emerging construct in the organizational behavior literature, and its incremental impact beyond individual job insecurity. The foremost implication is that organizations need to pay attention to the evolving climate perceptions about the future of jobs in the work environment, because such perceptions are related to critical employee behaviors.

## 1. Introduction

Job insecurity is among the most crucial work stressors affecting employees and organizations [1]. Past research indicates that job insecurity has detrimental consequences for several work- and health-related outcomes [2,3]. While job insecurity has always been a compelling topic for employees and organizations, the worldwide economic and labor market impacts of the coronavirus (COVID-19) pandemic [4] have made it a more pressing issue that needs to be examined thoroughly. Research has provided evidence that COVID-19 is associated with increased perceptions of job insecurity, which has implications for many consequences [5,6].

Job insecurity has been traditionally defined as the concerns or threats regarding the potential loss of the job itself (quantitative job insecurity) or loss of valued job features such as pay, promotion opportunities, working hours, and job content (qualitative job insecurity) [7,8]. It has recently been demonstrated that the construct is broader to also encompass the notion of job insecurity climate, that is, perceiving a climate concerning the risk of losing the job and/or valuable job features referred to as quantitative and qualitative job insecurity climate, respectively [9]. In contrast to the extensive research on individual job insecurity, only a few empirical studies have focused on job insecurity climate perceptions. Past research has, however, demonstrated that job insecurity climate has additional effects, beyond those of individual job insecurity, in explaining outcome variables, including health, demands, work–family conflict [9], job attitudes [10,11,12], and safety-related attitudes and behavior [13]. While research on job insecurity climate has focused on various outcomes, these outcomes have mainly concerned health and job attitudes and rarely included employee behaviors, per se.

Individuals may resort to various behavioral responses in the face of unsatisfactory working conditions. The exit, voice, loyalty, and neglect (EVLN) framework argues that leaving the organization (exit), trying to affect the organization in a desired way (voice), staying loyal to the organization (loyalty), or protesting by negligence (neglect) are responses employees may engage in when exposed to dissatisfying employment conditions (e.g., the prospect of losing a job) [14,15,16]. Several studies have supported the validity of the EVLN framework in explaining individual responses to challenging work situations [17,18,19]. Research has also shown that employees may resort to exit, voice, loyalty, and neglect in the face of perceived job insecurity [20,21,22,23]. However, in previous research, job insecurity perceptions were defined at the individual level rather than as climate perception.

Shoss [1] called for research to investigate the issue of job insecurity climate, arguing that the perceptions shared with others in the work environment may affect the attitudes and behaviors of individuals. Following Shoss’ call, the present study aimed to extend previous findings by examining the associations of job insecurity climate with EVLN behaviors. We had two research questions:

**Research Question 1**: How do the psychological collective climate perceptions about losing the job and valuable job features (quantitative and qualitative job insecurity climate, respectively) relate to employees’ exit, voice, loyalty, and neglect behaviors?

**Research Question 2**: Do job insecurity climate perceptions explain additional variance in these behaviors over individual job insecurity perceptions?

In the following sections, we first define the job insecurity climate. Then, we review the literature on how job insecurity (climate) relates to EVLN behaviors, concluding by stating the research hypotheses and presenting the conceptual model guiding the present study.

### 1.1. Job Insecurity Climate

The social information processing theory [24] emphasizes that social information provided by the social context shapes individuals’ needs, attitudes, and actions. It also affects their attentional processes, shaping their interpretations of environmental cues, and the evaluations of their needs. Based on this theory, it seems crucial to consider the role of the social context and the perceived climate surrounding individuals in order to better understand their reactions to the events.

In workplaces, individuals often work together in workgroups and teams. They share emotions, thoughts, and perceptions with others in those groups. Thus, the perception of job insecurity can be shared and exchanged through individual interactions within workgroups [11,12,13] and can become a psychological collective climate [9].

Scholars have conceptualized job insecurity climate as a psychological climate-related construct about losing a job and valuable job features [9,25]. Similar to the rationale for conceptualizing individual job insecurity as a stressor, researchers have viewed job insecurity climate as a contextual stressor with adverse impacts on employee work- and health-related outcomes. For example, Sora et al. [11] found that job insecurity climate was negatively associated with job satisfaction and organizational commitment after controlling for individual job insecurity. Låstad et al. [9] showed that job insecurity climate predicted work–family conflict, psychological distress, and poor self-rated health beyond individual job insecurity. Jiang and Probst [13] found that job insecurity climate was negatively associated with safety attitudes and behaviors. Recently, Hsieh and Kao [10] found that job insecurity climate had a negative indirect relationship with employees’ work engagement and job satisfaction through the perceived organizational obstruction. Although all these findings indicate that the climate of job insecurity, similar to individual job insecurity, is a phenomenon that has significant effects on various outcome variables, the current findings do not encompass the behaviors that employees consciously engage in. The EVLN framework includes behaviors potentially relevant to job insecurity climate perceptions.

### 1.2. Job Insecurity Climate and Employee Exit, Voice, Loyalty, and Neglect Behaviors

When employees face unfavorable conditions in their work situation, they may choose to leave the organization (exit), raise concerns and attempt to change the dissatisfying condition in a desired way (voice), stay and support the organization (loyalty), or become slack and engage in disregardful behaviors such as lateness and absenteeism (neglect) [14,26]. These four behaviors are defined into a two-dimensional model with constructive/destructive and active/passive dimensions. Voice and loyalty are the constructive responses in which individuals strive to keep or restore satisfactory employment conditions; exit and neglect are the destructive ones. Exit and voice are the active responses through which individuals take action to cope with dissatisfaction, while loyalty and neglect are the passive ones [16]. Exit is typically used by individuals who can easily find another job outside the organization, while employees who resort to voice typically have high status in the organization and trust that they will be listened to. In contrast, loyalty and neglect are responses primarily among lower-status employees, for whom exit and voice may not be realistic alternatives [14,26]. Numerous studies have supported this EVLN typology, composed of four distinct behavioral responses, both theoretically and empirically [27,28].

Past research has demonstrated significant associations between individual job insecurity and EVLN behaviors [20,22]. Regarding exit, the findings mainly indicate a positive association between individual job insecurity and exit, showing that, along with the job insecurity perception, employees’ propensity to leave the job or the organization increases [2,3]. Although the majority of previous research on this topic has focused on individual quantitative job insecurity, there is also support for a positive relationship between qualitative job insecurity and turnover intention. For example, Hellgren et al. [8] found that as employees’ perceptions of individual qualitative job insecurity increase, their turnover intention tends to increase. Beyond such a direct effect, Urbanaviciute and colleagues [29] showed that individually perceived qualitative job insecurity was indirectly related to turnover intention through the satisfaction of basic psychological needs for autonomy, competence, and relatedness. There is currently no finding demonstrating a link between job insecurity climate and exit behavior. Yet, it would be reasonable to expect that, similar to individually perceived concerns about losing the job and valuable job features, collective perceptions of job insecurity within a workgroup would be positively related to exit behavior.

Although contradictory results exist regarding the association between job insecurity and voice [22,23,30], many findings indicate a negative association between individual job insecurity and raising voice [20,31]. For example, Schreurs et al. [31] showed that individuals concerned about losing their job were less likely to speak up to authority. Likewise, Breevaart and colleagues [32] found that weekly feelings of individual quantitative job insecurity led to lower need fulfillment, which, in turn, lessened employee voice and raised employee silence in those weeks and thereafter. A similar pattern exists for individual qualitative job insecurity. Muñoz Medina et al. [33] showed that individual qualitative job insecurity was associated with lower levels of voice through a reduction in affective organizational commitment. Despite the literature suggesting an association between individual job insecurity and voice, the relationship between climatic job insecurity and voice has yet to be investigated. However, relying on the social exchange theory [34], it would be reasonable to argue that if the organization fails to provide job security, individually or collectively, employees’ tendency to behave favorably, including voice, might decrease. Thus, in the face of collective concerns about losing their job and deteriorating valuable job features, employees may be less inclined to raise their voices.

When it comes to loyalty, research has generally found a negative relationship between individual quantitative job insecurity and loyalty, suggesting that job insecurity is associated with decreased commitment to the organization [2,3,35]. Although most research on this topic has focused on the quantitative aspect of individual job insecurity, there are signs of a negative relationship also between qualitative job insecurity and loyalty. For example, Urbanaviciute et al. [36] found that individual qualitative job insecurity was related to lower organizational commitment, and this relationship was partially mediated by organizational control and employability. Moreover, past research on the relationship between job insecurity climate and loyalty provides some guidance for our study. More specifically, it has been found that job insecurity climate is negatively related to organizational commitment [11,12].

While research on the link between job insecurity and neglect is limited, past findings have yielded positive associations between organizational downsizing and the risk of long-term sick leave [37], as well as between job insecurity and avoidance behavior [38], suggesting that employees can engage in neglectful, avoidant, and disregardful behaviors in response to job insecurity. In addition, existing findings indicating a negative relationship between job insecurity climate and work involvement [12] suggest that the climate of job insecurity may be related to an increase in employees’ neglect behavior.

Based on previous research on individual job insecurity in relation to exit, voice, loyalty, and neglect—and the few studies on job insecurity climate—it would be reasonable to expect the relationship between job insecurity climate perceptions and such outcomes to be of a similar nature to those of individual job insecurity. To answer our research questions about how the climate of job insecurity relates to employee behaviors and whether such climate perceptions explain additional variance in these behaviors over individual job insecurity, we proposed the following hypotheses (see also Figure 1, which displays the study’s conceptual framework):

**Hypothesis 1.** 
*Quantitative job insecurity climate (psychological collective climate about losing the job) is related to higher levels of exit (H1a) and neglect (H1b), and lower levels of voice (H1c) and loyalty (H1d), when tested together with individual job insecurity dimensions.*


**Hypothesis 2.** 
*Qualitative job insecurity climate (psychological collective climate about losing valuable job features) is related to higher levels of exit (H2a) and neglect (H2b), and lower levels of voice (H2c) and loyalty (H2d), when tested together with individual job insecurity dimensions.*


**Hypothesis 3.** 
*Job insecurity climate perceptions explain additional variance in exit (H3a), voice (H3b), loyalty (H3c), and neglect (H3d) after considering the effects of individual job insecurity.*


## 2. Materials and Methods

### 2.1. Sample and Procedure

The cross-sectional study was conducted between May and September 2019 with an online survey using a convenience sampling method. The sample consisted of 245 employees working in different organizations in Turkey (51% women, mean age = 34, [range: 19–59], average organizational tenure = 5 [range: 1–27 years]). All participants were assured of confidentiality and informed that their answers would only be used for this research. No incentive was provided for participation. The missing values (1.14% of the data set) were imputed using the EM algorithm based on the recommendations by Tabachnick and Fidell [39]. Ethical approval for this study was acquired from the Human Subjects Ethics Committee of the Middle East Technical University, Turkey.

### 2.2. Measures

The items of all scales used in the study were translated into Turkish by a research team whose members had a good command of the English language. The back-translation from Turkish to English was then performed by a bilingual person unfamiliar with the item wordings in their original English version. After gathering the translated and back-translated information, the researchers created the proper Turkish translations for each item.

#### 2.2.1. Individual Job Insecurity

Individually perceived quantitative job insecurity was assessed using the three-item subscale, and qualitative job insecurity was measured using the four-item subscale of Hellgren et al. [8]. Sample items were “There is a risk that I will have to leave my present job in the year to come” for quantitative job insecurity (α = 0.64) and “My future career opportunities in [the organization] are favorable [reverse coded]” for qualitative job insecurity (α = 0.74). Items were rated on a seven-point Likert scale from 1 (strongly disagree) to 7 (strongly agree).

#### 2.2.2. Job Insecurity Climate

In this study, we operationalized job insecurity climate based on the referent-shift approach [40]. We asked individuals directly to report their job insecurity climate perceptions at an individual level instead of aggregating individual job insecurity ratings to obtain a measure of job security climate. The quantitative and qualitative dimensions of job insecurity climate were measured using the four-item subscales of Låstad et al. [9]. Sample items were “Many people are worried about losing their jobs at my workplace” for quantitative job insecurity climate (α = 0.88) and “Many people at my workplace express anxiety over their career development in the organization” for qualitative job insecurity climate (α = 0.82). Items were rated on a seven-point Likert scale from 1 (strongly disagree) to 7 (strongly agree).

We subjected the job insecurity items to confirmatory factor analysis to ensure that the four measures represented distinct dimensions of job insecurity. The results indicated that the four-factor structure of job insecurity—consisting of quantitative and qualitative individual job insecurity, and quantitative and qualitative job insecurity climate—provided a satisfactory fit to data (Satorra–Bentler χ^2^(84) = 188.62, *p* < 0.001, CFI = 0.96, RMSEA = 0.07, SRMR = 0.06). The four-factor representation clearly outperformed the model where all items were loaded on a single job insecurity factor (Satorra–Bentler χ^2^(90) = 746.79, *p* < 0.001, CFI = 0.74, RMSEA = 0.17, SRMR = 0.15), a two-factor model distinguishing between individual job insecurity and job insecurity climate (Satorra–Bentler χ^2^(89) = 735.25, *p* < 0.001, CFI = 0.75, RMSEA = 0.17, SRMR = 0.14), and a two-factor model distinguishing between quantitative and qualitative job insecurity (Satorra–Bentler χ^2^(89) = 415.75, *p* < 0.001, CFI = 0.87, RMSEA = 12, SRMR = 0.12), as well as a second-order model in which all four first-order factors loaded on a higher-order job insecurity factor (Satorra–Bentler χ^2^(86) = 217.08, *p* < 0.001, CFI = 0.95, RMSEA = 0.08, SRMR = 0.09).

#### 2.2.3. Exit, Voice, Loyalty, and Neglect

Exit, voice, loyalty, and neglect were assessed using related subscales of the EVLN scale of Hagedoorn et al. [41]. Participants were asked to point out the level to which they would apply the specified behaviors if they perceived job insecurity. Sample items were “Look for job advertisements in the newspapers to which you could apply” for the six-item exit measure (α = 0.86), “Try to work out an ideal solution in collaboration with your supervisor” for the eleven-item measure of considerate voice (α = 0.92), “Trust the organization to solve the problem without your help” for the five-item loyalty measure (α = 0.84), and “Come in late because you do not feel like working” for the five-item neglect measure (α = 0.83). Items were evaluated on a seven-point Likert scale ranging from 1 (definitely not) to 7 (definitely yes).

We conducted a confirmatory factor analysis on the items to ensure that the four measures represented distinct dimensions. The results revealed that the four-factor structure of EVLN (exit, voice, loyalty, and neglect) provided a satisfactory fit to data in the present sample (Satorra–Bentler χ^2^(293) = 668.85, *p* < 0.001, CFI = 0.94, RMSEA = 0.07, SRMR = 0.08).

#### 2.2.4. Demographic Variables

Participants were asked to report their age (in years), gender (1 = woman; 0 = man), education level (1 = four years university degree or more; 0 = lower education), organizational tenure (in years), and sector they were working in (1 = private sector; 0 = public sector).

### 2.3. Data Analysis

We performed hierarchical multiple regression analyses for each behavior (i.e., exit, voice, loyalty, and neglect) to examine the relation between the two job insecurity climate dimensions (quantitative and qualitative) and the four behaviors. The predictors were added in two pre-determined steps. In Step 1, we entered the demographic variables that had significant bivariate correlations with the outcome variables, and individual quantitative and qualitative job insecurity into the analyses to exclude their effects on the dependent variables. In Step 2, we added the quantitative and qualitative job insecurity climate dimensions to the model. These regression analyses enabled observing the specific variance explained by job insecurity climate dimensions in the four behaviors beyond the effects of the controlled demographic variables and individual job insecurity perceptions.

## 3. Results

Table 1 depicts the means, standard deviations, and correlations among the variables. Education and sector were significantly correlated with job insecurity dimensions and/or employee behaviors. More specifically, employees with higher education levels reported lower levels of quantitative job insecurity (individual and climate) and loyalty, and higher levels of exit than those with lower education levels. Private-sector employees reported higher quantitative job insecurity (individual and climate) and voice, and lower individual qualitative job insecurity than public-sector employees. Based on these bivariate associations, only education and sector were entered in the first step of the hierarchical multiple regression analyses, along with individual job insecurity dimensions, before adding the job insecurity climate dimensions in Step 2 to test the study hypotheses.

Table 2 reports the results of the hierarchical multiple regression analyses. The demographic variables (education and sector) and the two individual job insecurity dimensions entered in Step 1 explained 13%, 11%, 16%, and 4% of the variances in exit, voice, loyalty, and neglect, respectively. In Step 1, higher education was related to higher exit and lower loyalty, but unrelated to voice and neglect. Working in the private sector was associated with higher exit, but unrelated to voice, loyalty, and neglect. Individual quantitative job insecurity significantly predicted exit (β = 0.22, *p* < 0.01) and neglect (β = 0.18, *p* < 0.01), suggesting that participants who perceived threats to the continuity of their jobs reported higher preferences for quitting the job and protesting with neglect. Moreover, individual qualitative job insecurity was positively associated with exit (β = 0.20, *p* < 0.01) and negatively related with voice (β = −0.30, *p* < 0.001) and loyalty (β = −0.32, *p* < 0.001). These results indicate that participants who perceived threats of losing valued features of their jobs might be more prone to quit their jobs and less inclined to engage in voice and loyalty. In total, the first step accounted for 13% of the variance in exit, 11% in voice, and 16% in loyalty, while it did not explain a significant proportion of variance in neglect.

In Step 2, with the addition of quantitative and qualitative job insecurity climate dimensions to the model, all six variables explained 19%, 12%, 23%, and 6% of the variances in exit, voice, loyalty, and neglect, respectively. In this step, the effects of individual quantitative job insecurity in predicting exit and neglect were no longer significant. In addition, different from Step 1, working in the private sector was associated with lower loyalty.

Quantitative job insecurity climate was positively related to loyalty (β = 0.31, *p* < 0.001) and neglect (β = 0.22, *p* < 0.05), indicating that a psychological collective climate characterized by concern about losing the current job was associated with higher levels of loyalty to the organization and protesting with neglect. Hypothesis 1 stated that quantitative job insecurity climate is related to higher levels of exit (H1a) and neglect (H1b), and lower levels of voice (H1c) and loyalty (H1d) when tested together with individual job insecurity dimensions. Thus, Hypothesis H1b was supported. As quantitative job insecurity climate was not predictive of exit and voice, and it predicted loyalty in the opposite direction, H1a, H1c, and H1d were not supported.

Qualitative job insecurity climate was related to higher levels of exit (β = 0.20, *p* < 0.01) and lower levels of loyalty (β = −0.30, *p* < 0.001). These results suggest that those who experienced a climate of losing valuable job features were more likely to leave their jobs and less likely to be committed to their organizations. Hypothesis 2 stated that qualitative job insecurity climate is related to higher levels of exit (H2a) and neglect (H2b), and lower levels of voice (H2c) and loyalty (H2d), when tested together with individual job insecurity dimensions. Therefore, Hypotheses H2a and H2d were supported. However, as qualitative job insecurity climate did not predict voice and neglect, H2b and H2c did not receive support.

As shown in Table 2, the job insecurity climate dimensions entered in Step 2 accounted for significant increments in the explained variances for exit (ΔR^2^ = 0.06, ΔF = 8.29, *p* < 0.001) and loyalty (ΔR^2^ = 0.07, ΔF = 10.63, *p* < 0.001). These results indicate that job insecurity climate explained additional variance in employees’ exit and loyalty behaviors, beyond what was accounted for by individual job insecurity. Hypothesis 3 proposed that job insecurity climate explains additional variance in exit (H3a), voice (H3b), loyalty (H3c), and neglect (H3d) after considering the effects of individual job insecurity. Hence, the results provide support for H3a and H3c. However, the fact that the job insecurity climate dimensions did not account for significant increments in explained variance in voice and neglect means that we found no support for H3b and H3d.

## 4. Discussion

The current study investigated how quantitative and qualitative job insecurity climates were related to exit, voice, loyalty, and neglect behaviors among employees in Turkey, and whether such climate perceptions explained additional variance in these behaviors over individual job insecurity. Specifically, we hypothesized that quantitative job insecurity climate is related to higher levels of exit (H1a) and neglect (H1b) and lower levels of voice (H1c) and loyalty (H1d), and that qualitative job insecurity climate is related to higher levels of exit (H2a) and neglect (H2b), and lower levels of voice (H2c) and loyalty (H2d), when tested together with individual job insecurity dimensions (quantitative and qualitative). Our third hypothesis was related to the effect of job insecurity climate in explaining these behaviors beyond the effect of individual job insecurity, and stated that job insecurity climate explains additional variance in exit (H3a), voice (H3b), loyalty (H3c), and neglect (H3d) after considering the effects of individual job insecurity.

Regarding the associations between quantitative job insecurity climate and EVLN (Hypothesis 1), our study revealed that quantitative job insecurity climate was associated with higher levels of loyalty and neglect; however, it did not significantly predict exit and voice after controlling for the effects of individual job insecurity and the critical control variables (i.e., education and sector). These results support only H1b by suggesting that employees’ tendency to engage in negligent behaviors might be higher when they experience a collective concern about the continued existence of their jobs. On the other hand, the results did not support H1a, H1c, and H1d, as quantitative job insecurity climate was not predictive of exit and voice, and it predicted loyalty in the opposite direction (such that higher levels of quantitative job insecurity climate were associated with higher loyalty).

The supported positive association between job insecurity climate and neglect is in accordance with past findings indicating that threat of dismissal is associated with increased risk of long-term sick leave [37] and avoidance behavior [38], and decreased work involvement [12]. On the other hand, the non-significant relationships of quantitative job insecurity climate with exit and voice represent a different picture from previous findings in the job insecurity literature, which frequently reveals significant associations of individual quantitative job insecurity with exit [20,22] and voice [23,31]. The present findings thus suggest, contrary to what we hypothesized, that the mechanisms that prompt employees to quit their jobs or raise their voices in the face of the collective threat of job loss may differ from those in the case of individual job insecurity. One point to note is the possibility that some factors may fully mediate the relationships between quantitative job insecurity climate and exit and voice. Investigating potential mediating factors in these relationships would contribute to the literature.

In addition, our results unexpectedly showed that the climate of quantitative job insecurity was associated with increased loyalty. Although this finding challenges the widely held belief that job insecurity and loyalty are negatively related [2,3], it reminds us of Sverke and Hellgren’s [23] assertion that while reduced loyalty can be a natural outcome of job insecurity, increased loyalty may be evaluated as an effort to reduce uncertainty. In addition, the loyalty measure items, which appear somewhat reflective of feeling helpless and fatalism, might be an explication of the positive relation between quantitative job insecurity climate and loyalty.

Our results on the relationships between qualitative job insecurity climate and the EVLN behaviors (Hypothesis 2) revealed that, after controlling for the effects of individual job insecurity and the control variables (i.e., education and sector), qualitative job insecurity climate was related to higher levels of exit and lower levels of loyalty, but did not predict voice and neglect. These findings support H2a and H2d by suggesting that employees who experience a climate of losing valuable job features are more likely to leave their jobs and less likely to be loyal to their organizations. These supported hypotheses agree with previous findings demonstrating individual qualitative job insecurity to be associated with higher turnover intention [8,29] and lower organizational commitment [11,12,36].

Notably, our results—showing a non-significant association between exit and the quantitative aspect of job insecurity climate, but a positive relationship with the qualitative aspect—are in line with Hellgren et al.’s [8] finding that individual qualitative job insecurity has a stronger relationship with turnover intention than the quantitative aspect. This finding may suggest that threats to job features (individual and climate) may be more potent in predicting some outcome variables than threats to the jobs. Nevertheless, in our study, voice and neglect were not predicted by qualitative job insecurity climate. These results made us reject H2b and H2c by showing that, contrary to what we had expected, the experience of a climate of losing valuable job features was not a significant predictor of speaking up in the organization to make things better and engaging in negligent behaviors.

In our study, neither of the two job insecurity climate dimensions (quantitative and qualitative) significantly predicted voice. This result suggests that the psychological collective climate related to the future state of a job might not motivate employees to strive to change a problematic situation. Although this contradicts previous findings showing a significant relationship between individual job insecurity and voice [20,31], it may suggest that the perception of job insecurity, as an individual or collective experience, can differentiate its relationship with voice. Specifically, employees may have consciously chosen to remain silent in response to the collective concern regarding the future of their jobs. A lack of motivation to put one’s hands to the plough, or an evaluation of the cost of voice to be high and its anticipated effectiveness to be low, might be among the possible reasons behind this choice [42,43]. Surrounding factors such as economic conditions, labor legislation, conditions of trade unionism, and individual/collective cultural attitudes might have affected the choice of not to voice in the face of a job insecurity climate [44]. In addition, the voice measure we utilized in this study (i.e., considerate voice) could have influenced the results. With the awareness that there may be variations in the conceptualization and measurement of voice behavior [45], it should be noted that the type of voice behavior under consideration may affect the results regarding the relationship between job insecurity and voice. Whatever the reason, our findings regarding voice add to the contradictory findings regarding the job insecurity–voice relationship [42,43,44,46] and highlight the need for more research on this issue.

In terms of the additional variance explained by job insecurity climate in EVLN behaviors after controlling for the effect of individual job insecurity (Hypothesis 3), our study revealed that job insecurity climate explained additional variance in exit and loyalty, but not in voice and neglect, beyond what was accounted for by individual job insecurity (and the control variables education and sector), providing support for H3a and H3c. The additional explained variance in exit and loyalty of the job insecurity climate is in line with previous findings showing that the experience of job insecurity climate has been shown to predict employees’ health [9], job attitudes [10,11,12], and safety-related attitudes and behaviors [13], beyond the effect of individual job insecurity. Our results, as such, extend existing knowledge on the consequences of job insecurity climate and provide support for the social information processing perspective by showing that individuals’ reactions to events or circumstances at work cannot be understood independently of the influence of the social context [24].

### 4.1. Theoretical Implications

Despite some unexpected results (non-significant and opposite direction associations and non-significant additional variance of job insecurity climate in explaining some behaviors beyond the impact of individual job insecurity), the present study contributes to the literature in several ways. The first contribution of this study is that it responds to Shoss’ [1] call for empirical research examining job insecurity as a collective construct (i.e., job insecurity climate) by examining its associations with some critical behavioral consequences. Secondly, this study contributes to the literature by demonstrating the additional variance explained by job insecurity climate, over individual job insecurity, in behaviors that have not been addressed before in relation to job insecurity climate. All in all, our findings highlight the importance of taking a holistic approach to understand the consequences of job insecurity. In this study, we investigated how job insecurity climate related to EVLN behaviors and we did this by controlling for the effect of individual job insecurity (and critical demographic variables, namely education and sector).

As another contribution, the relationships investigated in this study were tested using data from a non-Western country, namely Turkey. Considering that research on job insecurity in relation to EVLN behaviors has typically been investigated using data collected from European countries [20,22,23,31], our findings contribute to the generalizability of extant research results regarding the consequences of job insecurity.

Taken together, our study extends the existing knowledge on the consequences of the job insecurity climate in organizational settings. Our findings should encourage future research to explore the dynamics of job insecurity climate perceptions.

### 4.2. Practical Implications

The results of our study point to some practical implications. Our results inform practitioners, managers, and policymakers that they need to be aware of the winds blowing in organizations while developing interventions to deal with a particular workplace stressor, namely job insecurity. It is crucial for practitioners to fully realize the form (individual or collective) and the nature (quantitative or qualitative) of job insecurity perceptions, perhaps by conducting needs assessments at regular intervals. The outcomes of job insecurity climate shown in this study (i.e., increased exit and neglect, and decreased loyalty) support the importance of allocating budgets for such preventive approaches in organizations. Moreover, beyond preventive practices, organizations should try to create a supportive climate for employees, and practitioners should develop adequate interventions in this regard. Such interventions may benefit from focusing on job insecurity as a collective stressor (job insecurity climate), which appears to have additional effects beyond those of individual job insecurity. To the best of our knowledge, no intervention has yet been developed for organizations in which a job insecurity climate prevails. Practitioners can work to determine whether various interventions designed for individual job insecurity perception also apply to job insecurity climate perceptions or to determine how such interventions can be modified to ameliorate the negative effects of an insecure climate.

### 4.3. Limitations and Direction for Future Research

Some limitations of this study need to be mentioned. First, due to our data’s cross-sectional nature, it is not possible to make causal inferences about the associations between job insecurity and employee behavioral reactions, and thus our results should be interpreted with caution. Future research using longitudinal data may consolidate the statistical strength of the information provided by this study [47]. Second, reliance on self-report measures may have increased the risk of overestimating the associations between job insecurity climate dimensions and EVLN behaviors because of common method variance [48,49]. We relied on self-reports as they may be the best and perhaps the only way to assess employees’ perceptions of job insecurity and EVLN behaviors. We are also aware of debates regarding the effect of common method variance, due to concerns related to its relevance in organizational field research [49]. A third limitation concerns the convenience/snowball sampling approach used to collect data. This sampling approach may have restricted the representativeness of our findings. Despite this constraint, we chose this data collection approach as it offers greater sample diversity, easier access, convenience, and lower costs and time investment [50]. Nonetheless, our findings await replication using random samples or population studies from other cultural contexts and organizational settings before any firm conclusions can be drawn regarding the possibility of generalizing our findings.

## 5. Conclusions

This study investigated how the psychological collective climate about losing a job and valuable job features (quantitative and qualitative job insecurity climate, respectively) relate to employees’ exit, voice, loyalty, and neglect behaviors, and whether job insecurity climate perceptions explain additional variance in these behaviors over individual job insecurity. We found quantitative job insecurity climate to be associated with higher levels of neglect (and higher levels of loyalty), and qualitative job insecurity climate to be related to higher levels of exit and lower levels of loyalty. These results, together with the finding that job insecurity climate explained additional variance in exit and loyalty beyond what was accounted for by individual job insecurity, demonstrate the importance of taking into account the psychological collective climate regarding job insecurity in explaining employee behaviors.

## Figures and Tables

**Figure 1 ijerph-20-05732-f001:**
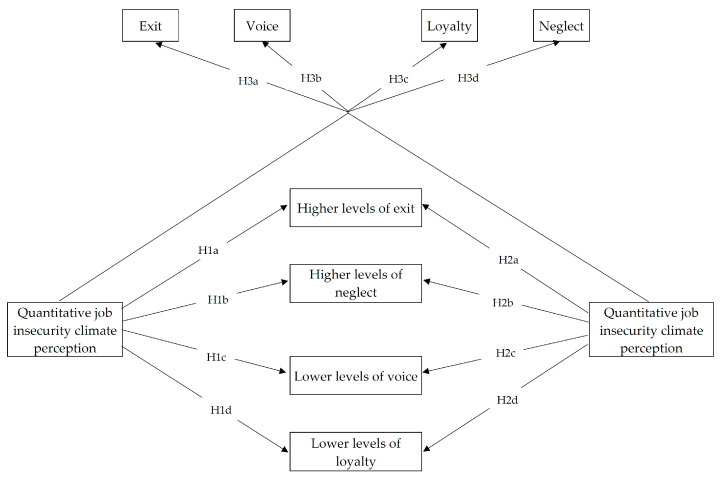
Conceptual model including study hypotheses. H3a-d indicates the expected incremental variance of job insecurity climate dimensions over the individual job insecurity dimensions.

**Table 1 ijerph-20-05732-t001:** Means, standard deviations, reliabilities (Cronbach’s alpha, in the diagonal), and bivariate correlations among the study variables.

	*M*	*SD*	1	2	3	4	5	6	7	8	9	10	11	12
1. Gender (woman)	0.51	0.50												
2. Age	33.81	8.08	–0.19 **											
3. Education (university)	0.70	0.46	0.16 *	–0.04										
4. Sector (private)	0.66	0.47	–0.12	–0.02	–0.31 ***									
5. Individual JI (quantitative)	2.89	1.43	0.03	–0.06	–0.18 **	0.17 **	(0.64)							
6. Individual JI (qualitative)	4.12	1.42	0.09	0.07	0.06	–0.27 ***	–0.02	(0.74)						
7. JI climate (quantitative)	3.00	1.61	–0.07	0.02	–0.17 **	0.26 ***	0.57 ***	–0.07	(0.88)					
8. JI climate (qualitative)	4.06	1.55	0.02	0.03	–0.01	–0.11	0.28 ***	0.27 ***	0.49 ***	(0.82)				
9. Exit	3.81	1.55	0.04	–0.13	0.15 *	0.12	0.20 **	0.15 *	0.27 ***	0.30 ***	(0.86)			
10. Voice	4.92	1.23	–0.04	0.06	–0.11	0.14*	–0.01	–0.32 ***	0.08	–0.00	–0.04	(0.92)		
11. Loyalty	3.85	1.36	–0.07	0.06	–0.25 ***	0.10	0.09	–0.31 ***	0.17 **	–0.20 **	–0.16 *	0.30 **	(0.84)	
12. Neglect	2.58	1.35	–0.10	–0.09	0.05	0.00	0.17 *	–0.04	0.19 **	0.03	0.30 ***	–0.16 *	–0.01	(0.83)

* *p* < 0.05, ** *p* < 0.01, *** *p* < 0.001 (N = 245). JI: Job insecurity. Scale range: Job insecurity dimensions, exit, voice, loyalty, and neglect were rated on 7-point Likert scales; gender: 0 = man, 1 = woman; age: in years; education: 0 = lower education, 1 = four years university degree or more; sector: 0 = public sector, 1 = private sector.

**Table 2 ijerph-20-05732-t002:** Results of hierarchical multiple regression analyses predicting exit, voice, loyalty, and neglect from JI (standardized regression coefficients).

	Exit		Voice		Loyalty		Neglect	
	Step 1	Step 2	Step 1	Step 2	Step 1	Step 2	Step 1	Step 2
Education (university)	0.24 ***	0.24 ***	–0.08	–0.08	–0.24 ***	–0.23 ***	0.08	0.08
Sector (private)	0.21 **	0.20 **	0.04	0.04	–0.07	–0.14 *	–0.02	–0.06
Individual JI (quantitative)	0.22 **	0.09	–0.03	–0.08	0.05	–0.03	0.18 **	0.10
Individual JI (qualitative)	0.20 **	0.15 *	–0.30 ***	–0.32 ***	–0.32 ***	–0.24 ***	–0.04	–0.01
JI climate (quantitative)		0.11		0.04		0.31 ***		0.22 *
JI climate (qualitative)		0.20 **		0.09		–0.30 ***		–0.11
Δ*R²*	0.13 ***	0.06 ***	0.11 ***	0.01	0.16 ***	0.07 ***	0.04	0.02
Model *R²*	0.13 ***	0.19 ***	0.11 ***	0.12 ***	0.16 ***	0.23 ***	0.04	0.06 *

* *p* < 0.05, ** *p* < 0.01, *** *p* < 0.001 (N = 245). JI: Job insecurity. Education: 0 = lower education, 1 = four years university degree or more; sector: 0 = public sector, 1 = private sector.

## Data Availability

The data presented in this study are not publicly available due to legal restrictions that guarantee the privacy of research participants. The data can be made available on request from the corresponding author.

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
