# Peer review of "Behavioral Reactions to Job Insecurity Climate Perceptions: Exit, Voice, Loyalty, and Neglect"

_ijerph, 2023, doi:10.3390/ijerph20095732_

Round 1

Reviewer 1 Report

I like to thank you for submitting a paper for the IJERPH.

The piece of research contained in this paper has been carefully thought through.  However, (I) there are no clear research questions, and (ii) it lacks a carefully designed conceptual model (where the hypothesis, variables and values are subsequently added)Whatever the article you are writing, the conclusion should "close" on the starting questions and if these are not well defined, how and about what do you conclude?

Please see the suggested Research questions and the Conceptual Model in my uploaded review file. 

Author Response

Answers to Reviewer 1:

  1. The piece of research contained in this paper has been carefully thought through. However, (I) there are no clear research questions, and (ii) it lacks a carefully designed conceptual model (where the hypothesis, variables and values are subsequently added). Whatever the article you are writing, the conclusion should "close" on the starting questions and if these are not well defined, how and about what do you conclude?

Answer: Thank you very much that you think our manuscript is carefully thought through. Moreover, thank you for pointing out the lack of research questions and a conceptual model in our manuscript and for providing templates for both the research questions and the conceptual model in the review file. In line with your suggestions, we have added research questions (page 2) before stating the study hypotheses and presented a figure showing the conceptual model of our study (page 5). We hope the text is more precise now and that you agree that the revised manuscript, including the research questions and the figure showing the conceptual model, is stronger.

  1. Please see the suggested Research questions and the Conceptual Model in my uploaded review file.

Answer: Thank you for providing us with the review file in which you specified your comments. We reviewed all the notes you mentioned in this file and tried to incorporate all the changes you suggested. We have revised the entire manuscript in terms of spelling errors and grammatical mistakes and made the necessary adjustments. Moreover, as you indicated in the review file, we clearly stated the quantitative and qualitative job insecurity climate definitions in parentheses when presenting our hypotheses. As you suggested, we added the research questions to the beginning of the introduction section (and reminded of them in the opening of the Discussion). We hope you agree with us that the manuscript has improved greatly and hope that the manuscript now is ready for publication in the IJERPH.

Reviewer 2 Report

The topic is interesting and very much relevant to the current context, moreover, the paper has wider theoretical and practical applications. The authors have put their best efforts to execute this paper. However, I have the following reservations and suggestions for the sake of improvement of the undertaken study:

1) The logical sequence of the abstract should be as 1) objectives, 2) methodology, 3) Findings, 4) conclusion and 5) implications. Thus, the authors should also rewrite the abstract in this sequence.

2) The authors did not establish the motivation, significance, and novelty of the undertaken study. The authors are suggested to improve this important factor in the "Introduction" section. The background of research should also be presented in the section. The structure of the review paper should also be presented in the end of Introduction section.

3) The literature should be presented in an audit form, and should be linked with the objectives of the current paper. The conceptual framework should be presented in the end of section “Literature Review”.

4) The conclusion should be added after the discussions section, conclusion is always one step ahead of findings. 

5) The practical, theoretical and societal implications should be discussed after the conclusion in separate headings, and in the light of the conclusion and discussions. 

6) Minor spelling and grammatical mistakes should be improved.

Author Response

Answers to Reviewer 2:

The topic is interesting and very much relevant to the current context, moreover, the paper has wider theoretical and practical applications. The authors have put their best efforts to execute this paper. However, I have the following reservations and suggestions for the sake of improvement of the undertaken study.

Answer: Thank you for finding our topic interesting, expressing that our study has theoretical and practical contributions, and appreciating our effort. In line with your recommendations and suggestions, we have made some changes to the manuscript. We hope you agree with us that the revised manuscript has improved after making the adjustments/changes you specified. 

  1. The logical sequence of the abstract should be as 1) objectives, 2) methodology, 3) findings, 4) conclusion and 5) implications. Thus, the authors should also rewrite the abstract in this sequence.

Answer: Thank you for pointing this out to us. We revised the abstract by making all specified parts (objectives, method, findings, etc.) more explicit so that readers can separate each section more clearly. In the abstract of the revised manuscript, the objectives, method, findings, conclusions, and implications sections are in the order you specified. We hope it reads better now.

  1. The authors did not establish the motivation, significance, and novelty of the undertaken study. The authors are suggested to improve this important factor in the "Introduction" section. The background of research should also be presented in the section. The structure of the review paper should also be presented in the end of Introduction section.

Answer: Thank you for this suggestion. We have reworked the introduction substantially and added a part at the end of the overview (page 2) specifying our research questions and how the remainder of the introduction is structured. Furthermore, we added a figure of our conceptual model that shows our research hypotheses (page 5). We hope the structure of our introduction section is clearer now and that the figure we added (along with the theoretical arguments) clarifies the motivation, significance, and novelty of our study.

  1. The literature should be presented in an audit form, and should be linked with the objectives of the current paper. The conceptual framework should be presented in the end of section “Literature Review”.

Answer: Thank you for pointing this out to us. In line with the suggestion of both you and Reviewer 1, we added a figure of our conceptual model showing research hypotheses at the end of the introduction section (page 5). We hope you agree that we can convey our research objectives more clearly by using this figure. The figure summarizes the arguments made in section 1.2 on the potential associations of job insecurity climate with exit, voice, loyalty, and neglect behaviors.

  1. The conclusion should be added after the discussions section, conclusion is always one step ahead of findings. 

Answer: Thank you for emphasizing this point. In the revised manuscript, we added a Conclusion section (5) after the Discussion (4), although the Conclusions typically is a sub-section of Discussion. In line with the other reviewer's suggestion, we repeated our research questions at the beginning of the Discussion before discussing the study findings. We also added sections on the contributions of the study (see below).

  1. The practical, theoretical and societal implications should be discussed after the conclusion in separate headings, and in the light of the conclusion and discussions. 

Answer: Thank you for this suggestion. In the revised manuscript, we added a section on theoretical implications (4.1, page 10-11) and practical implications (4.2, page 11), explaining how our findings can contribute to the existing literature and practical applications.

  1. Minor spelling and grammatical mistakes should be improved.

Answer: Thank you for pointing this out. We have revised the entire manuscript in terms of spelling errors and grammatical mistakes and have (hopefully) made the necessary adjustments. We hope the manuscript is appropriate in this regard now. Thank you very much for your recommendations, thoughts and ideas regarding our manuscript. We hope you agree with us that the manuscript has improved greatly and we hope that the manuscript now is ready for publication in the IJERPH.